# Engineering the Unicellular Alga *Phaeodactylum tricornutum* for Enhancing Carotenoid Production

**DOI:** 10.3390/antiox9080757

**Published:** 2020-08-16

**Authors:** Francesco Manfellotto, Giulio Rocco Stella, Angela Falciatore, Christophe Brunet, Maria Immacolata Ferrante

**Affiliations:** 1Stazione Zoologica Anton Dohrn, Villa Comunale, 80121 Naples, Italy; christophe.brunet@szn.it; 2Laboratory of Computational and Quantitative Biology, UMR 7238, Centre National de la Recherche Scientifique (CNRS), Sorbonne Université, Institut de Biologie Paris-Seine, F-75005 Paris, France; giulioste@gmail.com (G.R.S.); angela.falciatore@ibpc.fr (A.F.); 3Boston Consulting Group, Via Ugo Foscolo 1, 20121 Milano, Italy; 4Laboratory of Chloroplast Biology and Light Sensing in Microalgae, UMR 7141, Centre National de la Recherche Scientifique (CNRS), Sorbonne Université, Institut de Biologie Physico-Chimique, F-75005 Paris, France

**Keywords:** diatoms, *Phaeodacytlum tricornutum*, carotenoids, fucoxanthin, genetic engineering

## Abstract

Microalgae represent a promising resource for the production of beneficial natural compounds due to their richness in secondary metabolites and easy cultivation. Carotenoids feature among distinctive compounds of many microalgae, including diatoms, which owe their golden color to the xanthophyll fucoxanthin. Carotenoids have antioxidant, anti-obesity and anti-inflammatory properties, and there is a considerable market demand for these compounds. Here, with the aim to increase the carotenoid content in the model diatom *Phaeodactylum tricornutum*, we exploited genetic transformation to overexpress genes involved in the carotenoid biosynthetic pathway. We produced transgenic lines over-expressing simultaneously one, two or three carotenoid biosynthetic genes, and evaluated changes in pigment content with high-performance liquid chromatography. Two triple transformants over-expressing the genes Violaxanthin de-epoxidase (*Vde*), Vde-related (*Vdr*) and Zeaxanthin epoxidase 3 (*Zep3*) showed an accumulation of carotenoids, with an increase in the fucoxanthin content up to four fold. *Vde*, *Vdr* and *Zep3* mRNA and protein levels in the triple transformants were coherently increased. The exact role of these enzymes in the diatom carotenoid biosynthetic pathway is not completely elucidated nevertheless our strategy successfully modulated the carotenoid metabolism leading to an accumulation of valuable compounds, leading the way toward improved utilization of microalgae in the field of antioxidants.

## 1. Introduction

Diatoms, one of the major groups of microalgae, represent a potential source for commercial and industrial applications, because they naturally produce various beneficial substances for human activities, including foodstuffs and pharmaceuticals. Moreover, almost all of biomass can have a profitable use [1]. 

Diatoms have also been considered for applications such as production of biofuels, biofertilizers and nanomaterials, for industrial waste detoxification [2] and aquaculture feed [3]. Diatoms are rich in pharmaceutically active substances such as polyunsaturated fatty acids, vitamins, antioxidants, enzymes, polysaccharides and carotenoids [4]. Among diatoms, *Phaeodactylum tricornutum* is a very promising microorganism for application in industrial processes because it grows rapidly, doubling its biomass in a few hours [1]. Its growth is cost-effective, sustainable and can be easily controlled in indoor and outdoor conditions [5]. *P. tricornutum* is also the best-established diatom molecular model system, with cutting edge tools in place to alter gene expression in transformed cells by genetic over-expression, gene silencing and genome editing [6]. 

We focused our work on carotenoid production enhancement in *P. tricornutum*. Bioactivity and human health benefits of carotenoids are increasingly described [7].

Carotenoids are tetraterpenoids formed by a linear C40 main chain with multiple conjugated double bonds [8]. In diatoms, as in all photosynthetic organisms, carotenoids are synthesized via the methylerythritol phosphate (MEP) pathway [9,10]. Carotenoids associate with clorophyll to form the pigment-protein complexes of the photosinthetic apparatus [11]. 

Diatoms are phylogenetically closer to brown algae than to the green lineage [6] and their carotenoid profile is different from that found in plants and green algae: fucoxanthin is the most abundant carotenoid followed by diadinoxanthin (Dd), diatoxanthin (Dt) and β-carotene. Chlorophyll *a* and chlorophyll *c* form with fucoxanthin the fucoxanthin-chlorophyll protein complex that performs light-harvesting. Dx is known to play an important role in photoprotection [12,13], belonging to the so-called xanthophyll cycle: the diatoxanthin–diadinoxanthin cycle (Dd-Dt). Diatoms also present the more widely distributed violaxanthin–zeaxanthin cycle. 

Fucoxantin presents anti-obesity, anti-diabetic and anti-cancer properties. Fucoxanthin and its metabolite, fucoxanthinol, are well established as strong antioxidant compounds [14], they have radical scavenging activities which are 13.5 and 1.7 times higher, respectively, than the activity of α-tocopherol, and fucoxanthin anticancer effect is stronger than that of β-carotene [15]. Dd prevents damage resulting from exposure of skin and hair to the UV and visible range of the solar spectrum [16,17]. 

The biosynthetic carotenoid pigments pathway has been extensively studied in plants and green algae; however, diatoms show different metabolic features compared to plants [18] and, as mentioned, use unique pigments, that are not present in other species, for light harvesting and photoprotection [19]. 

The biosynthetic carotenoid pathway is still not completely understood, and reactions and enzymes from violaxanthin to Dx and fucoxanthin are still hypothetical (Figure 1). 

Attempts to overexpress the 1-deoxy-d-xylulose 5-phosphate synthase (*Dxs*), the gateway enzyme of the terpenoid pathway, or the Phytoene synthase (*Psy*), controlling the initial step of the carotenoid biosynthesis, resulted in a moderate increase in fucoxanthin content [22,23]. 

In the *P. tricornutum* genome there are seven genes putatively involved in violaxanthin-zeaxanthin and Dd-Dt cycles, four for the de-epoxidase reactions (Violaxanthin de-epoxidase *Vde*, *Vde*-like *Vdl1* and 2, and the *Vde*-related, *Vdr*), which convert violaxanthin in zeaxanthin or Dd in Dt, and three for the reverse reactions (Zeaxanthin epoxidases, *Zep1*, *2* and *3*) [24]. The expansion of these gene families is in contrast with what happens in plants, where only one *Vde*, one *Vdr* and one *Zep* are present. Of the three *Zeps* genes, *Zep1* was found to be non-functional in zeaxanthin epoxidation whereas *Zep2* and *Zep3* were able to restore zeaxanthin epoxidation and a functional xanthophyll cycle in *Arabidopsis thaliana* [25]. However, these enzymes exhibited different catalytic activities; for example, ZEP2 exhibited a broader substrate specificity with respect to ZEP3, leading to the hypothesis that this enzyme could be involved in the Dd-Dt cycle [25]. De-epoxidation of the violaxanthin cycle in chlorophytes is catalyzed by VDE. Acting in both xanthophyll cycles, the diatom VDE has been shown to participate in high light acclimation and non-photochemical quenching (NPQ) kinetics [26,27]. *Vde* gene expression is related to photoprotection and strongly induced by high light stress. In vitro, the VDE enzyme can use both violaxanthin and Dd as substrate for de-epoxidation reaction [28,29]. Lavaud et al. reported that *Vde* knock-down lines negatively impact de-epoxidation reactions and present Dd-Dt accumulation [27]. The function of the VDL and VDR proteins remains unknown. They probably have a xanthophyll cycle activity in addition to VDE, perhaps differing in localization and functional role as indicated by the differential light induced expression of the *Vdl* compared to the *Vde* genes [9,24,27]. *Vdr* is a high light-induced gene and appears to be generally present in all chlorophytes [24]; it supposedly participates in de-epoxidation reactions in addition to VDE at a certain threshold of light and lumen acidification [30]. VDL2 has been recently overexpressed in the diatom *Thalassiosira pseudonana*, leading to an increase in fucoxanthin and a reduction in Dd-Dt, without a net change in their sum nor in β-carotene, suggesting that VDL2 is required in the step leading from the Dd-Dt pigments to fucoxanthin [31]. Reactions from violaxanthin to fucoxanthin and enzymes involved are still unknown, but it had been hypothesized that the reactions are catalysed by VDLs proteins. VDE, VDL1, and VDL2 from *P. tricornutum* have been recently expressed in *Escherichia coli*, and tested in an in vitro assay commonly used for measuring VDE activity. VDE showed the expected de-epoxidase activity by converting both violaxanthin to zeaxanthin and Dd to Dt, VDL2 showed no catalytic activity, whereas VDL1 converted violaxanthin to neoxanthin, suggesting that violaxanthin is its major native substrate [32]. 

Efforts are needed in order to target the rate limiting steps in the pathway: a strategy in which more than one enzyme is overexpressed in the same strain might be more effective, similar to what has been shown for example in yeast, where the simultaneous overexpression of two genes involved in astaxanthin production led to more product than overexpression of the single genes [33]. 

Our aim was to enhance pigments production in *P. tricornutum*. Since the exact function of each enzyme of the xanthophyll cycle is still unclear, to maximize chances of interfering with the biosynthetic mechanisms, in parallel with single overexpression strategies already applied in past studies, we simultaneously tested different double and triple combinations of VDEs and ZEPs genes using biolistic transformation.

## 2. Materials and Methods

### 2.1. Algal Cultures

The *P. tricornutum* Pt1 strain (Pt1 8.6 CCMP2561) was used in this study. Wild type cells were grown at 18 °C, in a 12 h light/12 h dark photoperiod using white fluorescence neon lamps (Philips TL-D 90), at irradiance of 90 μmol m^−2^ s^−1^ in ventilated flasks in f/2 medium (Guillard, 1975) [34]. Transgenic cells were grown under the same condition in selective liquid f/2 medium supplemented with 50 µg/mL zeocin.

### 2.2. Plasmids Construction

Total RNA was isolated from *P. tricornutum* cells using TriPure Isolation Reagent (Sigma-Aldrich, Saint Louise, MO, USA) following manufacturer specifications. Then, cDNA was synthesized with a QuantiTect Reverse Transcription Kit (Qiagen, Hilden, Germany) and was used as template for the amplification of the transcripts of interest. Transformation vectors for VDEs and ZEPs (Table 1) overexpression were generated by amplifying the cDNA using the primers described in Appendix A. DNA fragments amplification was performed using Q5^®^ High-Fidelity DNA Polymerase (New England Biolabs, Ipswich, MA, USA) according to the manufacturer’s instructions. The full-length cDNA sequences were cloned in an Invitrogen Entry vector (pENTR) and then in the diatom pKSFcpBpAt-C-3HA destination vector, with the Gateway technology [35].

The plasmids containing the *Pds*, *Lcy*, *Psy* and *Lut1* genes were synthesized from cDNA using the primers described in Appendix A. DNA fragments amplification was performed using Q5^®^ High-Fidelity DNA Polymerase (New England Biolabs, Ipswich, MA, USA) according to the manufacturer’s instructions. PCR amplification was performed using a double annealing temperature, first ten cycles at the annealing temperature of the primers without tails and then 25 cycles at the annealing temperature of the entire primer sequence. The amplicons were purified by agarose gel electrophoresis using a QIAquick^®^ Gel Extraction Kit (Qiagen, Hilden, Germany). PCR products and plasmids were digested with the restriction enzyme EcoRI (New England Biolabs, Ipswich, MA, USA). The digested amplicons and plasmids pfcpb-OX [36] were ligated using the T4 DNA Ligase (New England Biolabs, Ipswich, MA, USA) according to the manufacturer’s instructions. We proceeded cloning plasmids in TOP10 competent cells (Thermo Fisher Scientific, Waltham, MA, USA). Plasmids were purified using QIAprep Miniprep Kit (Qiagen, Hilden, Germany) after overnight growth. All purified plasmids were further verified by Sanger sequencing.

### 2.3. Microparticle Bombardment

The plasmids were introduced into *P. tricornutum* cells using the Biolistic PDS-1000/He particle delivery system (Bio-Rad Laboratories, Hercules, CA, USA). We co-transformed *P. tricornutum* cells using the pFCPFp-Sh-ble vector conferring zeocin resistance and the carotenoid biosynthesis genes as described in Falciatore et al. (1999) [36]. For single gene transformation, we coated tungsten particles with 3 μg of resistance vector + 3 μg of the transformation vector. For double gene transformation, we coated tungsten particles with 2 μg of resistance vector + 2 μg for each transformation vector. For triple gene transformation we coated tungsten particles with 1.5 μg for each transformation vector. 

### 2.4. Positive Colony Screening 

PCR screening to select positive transformants colony was performed on cell lysate as described in Falciatore et al. (1999) [36]. To screen positives transformants, we designed a forward primer annealing to the exogenous fcpB sequence and specific reverse primers, annealing to the carotenoid biosynthesis genes (Figure 2 and Appendix A).

### 2.5. Pigment Profile

Pigment profile was analyzed following the protocol of Smerilli et al. (2017) [37]. Samples of approximately 15 × 10^7^ cells of *P. tricornutum* were harvested during the exponential growth phase and immediately filtered on 25-mm GF/F glass-fiber filters (Whatman™, Whatman International Ltd., Maidstone, UK). Pigments were extracted with 100% methanol by mechanical grounding, then the homogenate was filtered onto Whatman 25 mm GF/F filters. A Hewlett Packard series 1100 HPLC (Hewlett Packard, Wilmington, NC, USA), equipped with a C8Kinetex column (2.6 µm diameter; 50 mm × 4.6 mm; Phenomenex^®^, Torrance, CA, USA) was used to separate pigments, which were thus detected spectrophotometrically at 440 nm using a Hewlett Packard photodiode array detector, model DAD series1100. Determination and quantification of pigments were performed using pure pigments (D.H.I. Water & Environment; Horsholm, Denmark).

### 2.6. Gene Expression Profile 

The expression levels of target genes were evaluated by qPCR. We designed couples of primers to amplify the endogenous and exogenous genes (Appendix A). Total RNA was isolated from approximately 5 × 10^7^ transformed cells using TriPure Isolation Reagent (Sigma-Aldrich, Saint Louis, MO, USA). Reverse transcription (RT) was performed using QuantiTect Reverse Transcription Kit (Qiagen, Hilden, Germany). The qPCR experiments were performed in triplicate in a ViiA 7 Real-Time PCR System (Applied Biosystems, Foster City, CA, USA) using Fast SYBR™ Green Master Mix (Applied Biosystems, Foster City, CA, USA), following manufacturer instructions. The reference genes used were *H4* and *RPS* [35]. Quantification were made following the Δ-Δ-Ct method. The results are mean ± SD of at least three separate experiments, measuring each parameter by triplicate (*n* = 3).

### 2.7. Western Blot Analysis

Protein extraction was performed from 5 × 10^8^
*P. tricornutum* cells at the exponential growth phase, collected by centrifugation at 2000× *g*, following the published “TCA protein extraction from diatoms” (dx.doi.org/10.17504/protocols.io.bc7rizm6) [38]. Then, 40 ng of total proteins were loaded on 10% polyacrylamide gels and resolved by SDS-PAGE. Proteins were transferred to a nitrocellulose membrane using a Trans-Blot Semi-Dry Electrophoretic TransferCell (Bio-Rad Laboratories, Hercules, CA, USA) for 1 h at 140 mA in Towbin buffer. Transferred proteins were visualized by Ponceau S staining. Blocking was performed in PBS-T buffer supplemented with 5% defatted milk for 1 h at room temperature. A rabbit anti-PsbD antibody (Agrisera, Vännäs, Sweden) and a rabbit anti-HA antibody (Agrisera, Vännäs, Sweden) were used for detection of control D2 protein of PSII and HA tagged exogenous proteins, at 1:20,000 dilution in blocking buffer for 1 h. Incubation with the horseradish peroxidase-coupled secondary anti rabbit IgG antibody (Agrisera, Vännäs, Sweden), diluted 1:40,000, lasted for 1 h. Chemoluminescence was detected using the ECL Western Blotting Substrate (Thermo Fisher Scientific, Waltham, MA, USA). Membranes were imaged with a Gel Doc XR imaging system (Bio-Rad Laboratories, Hercules, CA, USA) to quantify band intensities by densitometry, using Quantity-One software (Bio-Rad Laboratories, Hercules, CA, USA).

### 2.8. Non-Photochemical Quenching of Chlorophyll Fluorescence (NPQ)

Fifteen min. dark-acclimated samples were inserted in a DUAL-PAM (Heinz Walz, Effeltrich, Germany) to estimate NPQ (non-photochemical quenching) [37]. The actinic light was setup at 480 μmol photons m^−2^ s^−1^ lasting 10 min., and the maximum fluorescence yield was estimated every min. Fm was obtained applying a saturating pulse of red light (4000 μmol m^−2^ s^−1^, lasting 450 ms). NPQ was quantified using the Stern-Volmer expression [37]. 

## 3. Results

With the aim to increase carotenoids production, we transformed *P. tricornutum* cells with single or multiple combinations of plasmids for overexpression of genes putatively involved in the pigments biosynthetic pathway (Figure 1, Appendix A). We obtained over 400 resistant colonies. After isolation, we PCR-screened the resistant colonies to confirm exogenous transgenes integration, using a forward primer annealing on the inserted FcpB promoter and a reverse primer annealing on the gene coding sequence in order to discriminate exogenous genes integration (Appendix A). 

The effect of single and multiple genes integration on the carotenoid content under light irradiance of 90 μmol m^−2^ s^−1^ was evaluated by HPLC analysis, in all those cultures which appeared healthy and grew normally (Appendix A). In this first pass large screening, where it was not feasible to grow each culture in triplicate, we tested each sample once to identify the most promising strains. For all the single overexpressing (OE) strains and for most part of the double OE strains the carotenoid content was not substantially different from the content in the wild type strain. From the HPLC screening, one of the double transformants (*Vdr/Vde*) and four triple transformant lines (*Vdr/Vde/Zep3*) showed a marked increase in the pigment content (Appendix A). 

We therefore focused our attention on strains T2 and T3, which showed the strongest changes (Appendix A). In order to confirm the initial results, we repeated HPLC measurements different times and in triplicate (Appendix A) for these two strains as well as for strain T1, which had integrated the same genes (Figure 3a–c) but did not show any alteration in its pigment content in the first pass screening and that we used as a negative control in subsequent analyses (Figure 3d and Table 2).

Transformants strains were analysed by quantitative RT-PCR (qPCR). We confirmed the positive correlation between carotenoid amount variations and exogenous gene expression. In particular, *Zep3*, *Vde* and *Vdr* were upregulated between 2 and 3 folds in strains T2 and T3 with respect to the wild type, as shown in Figure 4a, while T1 did not show significant variations. To confirm the overexpression of the biosynthetic enzymes, we looked at the presence of exogenous proteins by Western blot analysis. Using an anti HA antibody, we detected the exogenous VDR, ZEP3 and VDE HA-tagged proteins of a predicted molecular weight of 84 kDa, 71 kDa and 66 kDa, respectively, in the multiple transgenic lines T1, T2 and T3, albeit with different intensities. The D2 protein of PSII (39 kDa) was used as control for normalization. Strains T2 and T3 showed a stronger expression of exogenous proteins with respect to the T1 strain (Figure 4b,c).

To evaluate whether the increased pigment content led to increased photoprotection, we measured the non-photochemical quenching (NPQ) capacity between transformants and wild type strains and could not detect any significant difference in the values obtained (Figure 5). We did not notice any obvious difference compared to the wild type as far as growth, colour or morphology. 

## 4. Discussion

Natural bioactive compounds to be used in cosmeceuticals, pharmaceuticals or nutraceuticals are required for preserving human health and wellness. Among the most bioactive relevant families of compounds, carotenoids are known to be effective in promoting human health [39]. Marine microalgae are one of the most promising natural sources of bioactive compounds, such as carotenoids [40], often displaying fast growth, high bioactive compounds content, strong resistance and suitability for large-scale cultivation. In this context, diatoms are interesting models. However, to address the market challenges, productive costs need to be contained, enhancing the productivity yield of such compounds from microalgal biomass. 

The knowledge acquired by sequencing microalgal genomes allows us to predict the metabolic pathways by mapping annotated genes to the database resources, driving the selection of suitable target genes for genetic engineering applications [41]. In recent years, different research has aimed to enhance carotenoids production in the model diatom *P. tricornutum* overexpressing a single gene in the pigment’s biosynthetic pathway and/or optimizing culture conditions. In one study, overexpression of the *Psy* gene in *P. tricornutum* increased fucoxanthin content approximately 1.45 fold with respect to the wild type, although the content of other pigments did not increase [22]. In a similar study, overexpression of *Dxs* led to an increase in fucoxanthin of 2.4 fold, while overexpression of *Psy* led to a 1.8-fold increase [22,23]. Different studies using high-silicate medium and modulating red–blue light with an irradiance of 128–200–255 μmol m^−2^ s^−1^ also affected the fucoxanthin content [42]. 

The steps from zeaxanthin to violaxanthin and from Dd to Dt are catalysed by the ZEPs and VDEs gene family members (Figure 1) and are thought to be key players in the carotenoid biosynthetic pathway and found in all photosynthetic organisms. In the present study, by overexpressing *Vde, Vdr* and *Zep3* genes simultaneously in *P. tricornutum*, we achieved a correlated increase in the amounts of transgenic mRNA, exogenous proteins and carotenoids. Two independent transgenic strains showed a relevant increase in the carotenoids amount, up to a 4-fold increase in fucoxanthin, up to 3-fold in Dd and Dt, and up to 2-fold in the β-carotene content with respect to the wild type. Independent HPLC measurements for these strains taken at different times confirmed the pigment increase, indicating that the phenotype is stable. The increased Dt-Dd in the triple transgenic strains did not induce greater NPQ development with respect to the wild type, probably meaning that the increased Dt molecules are not conjugated to photo-antenna proteins, as already observed under peculiar light conditions in another diatom [43]. 

These two strains with increased pigment content harbour a combination of genes which are modulated in the light/dark cycle and involved in light acclimation [30,44,45,46]. In 2020, Kuczynska et al. found that cultures grown under an irradiance of 30 μmol m^−2^ s^−1^ and subsequently subjected to light stress with an irradiance of 700 μmol m^−2^ s^−1^ displayed a strong expression of *Zep3* and *Vde* genes during the initial days of light adaptation [44]. In 2009, Nymark et al. studied cells acclimated to 35 μmol m^−2^ s^−1^ irradiance and later exposed to 500 m^−2^ s^−1^ irradiance, and found an up-regulation of *Vdr* (after 12 h exposure) and *Zep3* (after 0.5 h exposure), while *Zep1* was down-regulated, suggesting that these latter two genes encode enzymes with different functions [30]. It has been shown that VDE can use Dd or violaxanthin as a substrate [27,29], while for VDR and ZEP3 the exact role and the preferred substrates in *P. tricornutum* have not been defined but possibly *Zep3* encodes the enzyme that converts Dd to Dt in response to light stress [30]. We note that the highest pigment content is detected in the strain with the highest VDR induction, and that a moderate increase in carotenoids was also observed for a double transformant harbouring *Vdr* and *Vde* but not *Zep3* (Appendix A), indicating that a large part of the phenotype could be linked to the overexpression of the two former enzymes. *Vdr*, *Vde* and *Zep3* genes expression is related to light-dark cycle, in particular, *Zep3* is robustly expressed during the light phase and under light stress, while *Vdr* and *Vde* are normally present at low levels and become upregulated under light stress [30,44,45,46]. *Zep3* OE may not be influent if the basal level of the enzyme is already very high. Contrastingly, the ectopic OE of enzymes normally not strongly expressed may induce a perturbation in the final steps of the pathway and maybe also affect regulatory loops that still remain unknown. Other transgenic lines revealed modification in the pigment content with a trend towards accumulating more (Appendix A), although replicate HPLC measurements would be needed to confirm the phenotype in those lines. The number of insertions and the genome insertion site/s for each transgene is not controllable with the methodology chosen for transformation and can vary, leading to variable levels of transcripts in each different OE strain. More detailed examinations of the mRNA and protein content in each transformant could help in better understanding the basis for the changes that we observed in T2 and T3, although this is beyond the scope of the current work.

Here, the increase in the different carotenoids represents an important goal, since the complementarity and potential synergy of their bioactivity might enhance the human health benefit of such biomass. Dd, Dt and fucoxanthin display high antioxidant activity [14,47]. Dd and Dt are known to present a great capacity of scavenging [48], while Dt also displays antiproliferative capacity against cancer cells [43]. Moreover, fucoxanthin is known to have anti-obesity and anticancer activities [39] while β-carotene is an important pro-vitamin A. Moreover, the growth of the triple transgenic strain is not limited or impaired compared to the wild type under moderate light conditions, meaning that we succeed in enhancing the final harvesting of bioactive carotenoids increasing their content per unit of biomass without lowering the total biomass production. This is a crucial aspect for industrial scale-up, where energy used for growing is an important limiting factor.

More in-depth analyses will be needed to reveal whether other growth conditions can enhance the pigment content phenotype; for instance, by using different light intensities or by subjecting cultures to light stress, and whether the presence of more pigments can lead to other alterations in the cell physiology. 

## 5. Conclusions

Our strategy, in which multiple transgenes have been expressed simultaneously in *P. tricornutum* cells, has successfully led to the generation of transgenic strains displaying an increased content of commercially valuable pigments. Future studies should focus on scaling up growth, to extract and test the pigments produced by the triple *Vdr/Vde/Zep3* OE strains for nutraceutics and similar applications. Dedicated investigations with a thorough phenotypic analysis will also contribute to refining our knowledge of the role of each enzyme in the diatom carotenoid biosynthetic pathway. This study lays the foundation for implementing genetic engineering aiming to enhance the role of microalgae for biotechnological applications, and represents a step towards using this microalga as a commercially sustainable source of these high valuable compounds.

## Figures and Tables

**Figure 1 antioxidants-09-00757-f001:**
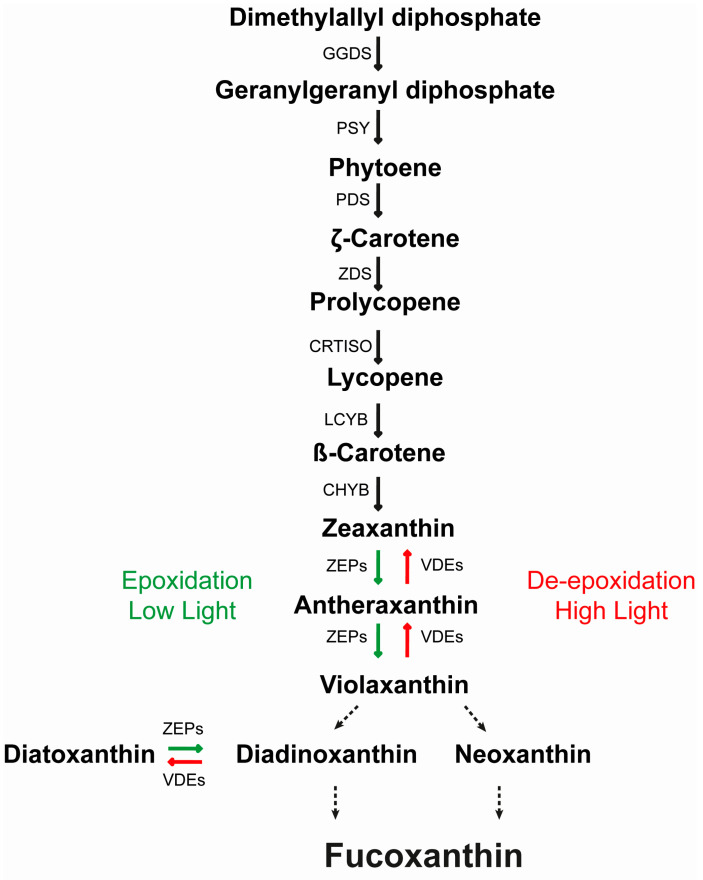
Schematic representation of the carotenoid biosynthesis pathway in *P. tricornutum*, according to the hypothesized pathways proposed by Lohr and Wilhelm 1999 [20] and Dambek et al. 2012 [21]. Dashed lines are used for unknown enzymes. Red arrows represent de-epoxidation reactions and green arrows represent epoxidation reactions operating in the two xanthophyll cycles. GGDS: geranylgeranyl diphosphate synthase; PSY: phytoene synthase; PDS: phytoene desaturase; ZDS: ζ-carotene desaturase; CRTISO: carotene cis/trans isomerase, prolycopene isomerase; LCYB: lycopene cyclase b; LUT: lutein deficient-like; ZEP: zeaxanthin epoxidase; VDE: violaxanthin de-epoxidase. Adapted from M. Bertand et al. (2010) and P. Kuczynska et al. (2015) [9,19].

**Figure 2 antioxidants-09-00757-f002:**
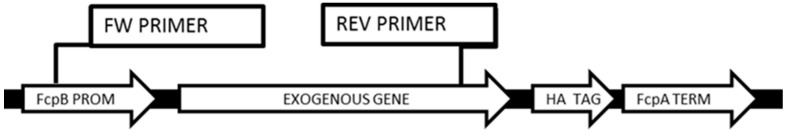
Schematic representation of the construct and position of the primers annealing sites used to screen exogenous gene integration.

**Figure 3 antioxidants-09-00757-f003:**
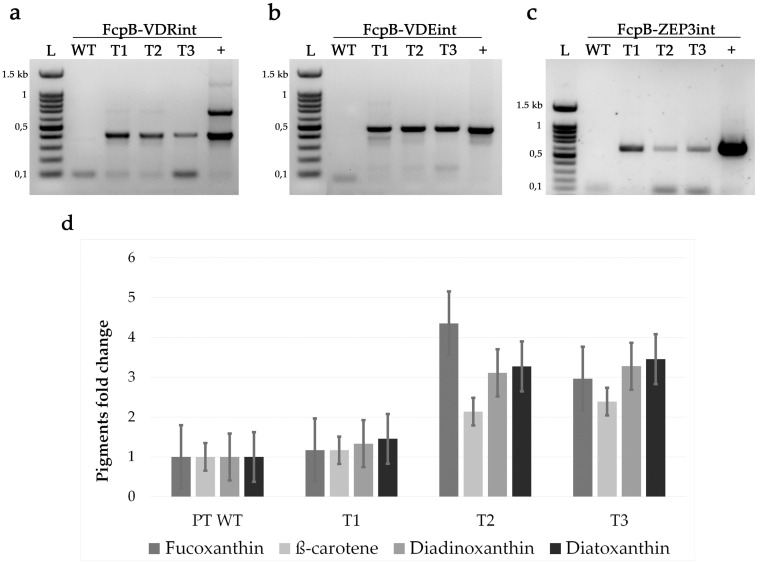
Transgene integration and pigments change in *P. tricornutum* wild type and OE strains T1, T2 and T3. (**a**); PCR screening for *Vdr*, using Forward FcpB and reverse VDRint primers, expected fragment of 383 bp. (**b**); PCR screening for *Vde*, using Forward FcpB and reverse VDEint primers, expected fragment of 443 bp. (**c**); PCR screening for *Zep3*, using Forward FcpB and reverse ZEP3int primers, expected fragment of 848 bp. L; 100 bp ladder. PT WT; *P. tricornutum* wild type. T1, T2. T3, triple transformants. +, positive control, the vector used for transformation. (**d**); Fold change in the pigments content normalised for chlorophyll *a* in transgenic strains with respect to wild type. Data represent mean (*n* = 3) and SD.

**Figure 4 antioxidants-09-00757-f004:**
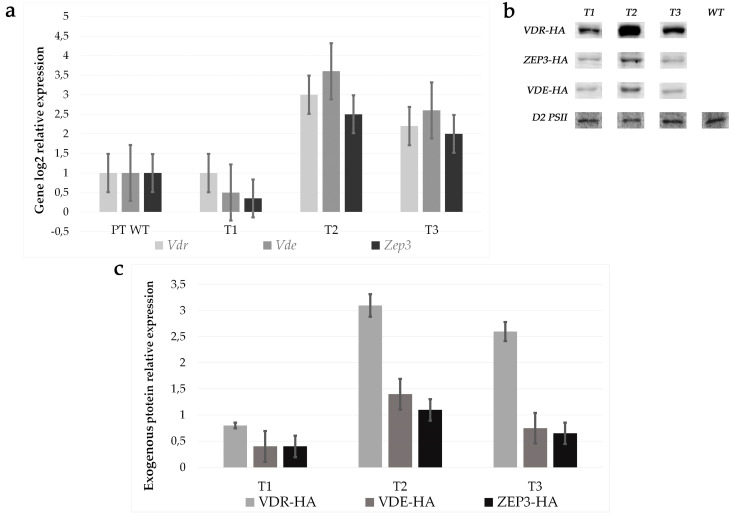
Transcripts and HA tagged proteins relative expression in *P. tricornutum* wild type and OE strains T1, T2 and T3. (**a**), Relative gene expression (endogenous and exogenous) as assessed by qPCR. (**b**), Western blot analysis performed on PT WT, T1, T2 and T3 in order to detect exogenous HA tagged proteins and control protein D2 of PSII. (**c**), relative exogenous protein expression.

**Figure 5 antioxidants-09-00757-f005:**
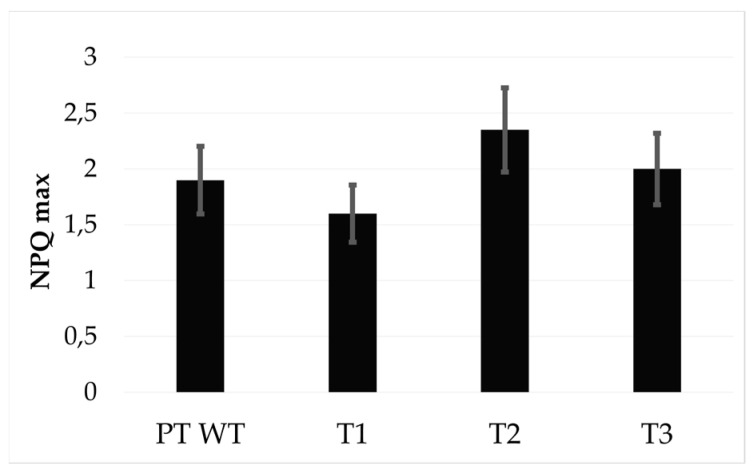
Non-photochemical quenching (NPQ max) developed by the four strains analysed.

**Table 1 antioxidants-09-00757-t001:** Genes selected for overexpression vectors construction.

Gene Description	Gene Name	Gene ID
Violaxanthin deepoxidase like protein	*Vde*	Phatr3_J51703
Violaxanthin deepoxidase-like 1 protein	*Vdl 1*	Phatr3_J36048
Violaxanthin deepoxidase-like 2 protein	*Vdl 2*	Phatr3_J45846
Zeaxanthin epoxidase like protein	*Zep 1*	Phatr3_J45845
Zeaxanthin epoxidase2-like protein	*Zep 2*	Phatr3_J5928
Zeaxanthin epoxidase3-like protein	*Zep 3*	Phatr3_J10970
Violaxanthin deepoxidase-related protein	*Vdr*	Phatr3_J43240
Phytoene desaturase	*Pds*	Phatr3_J10438
Lycopene beta cyclase	*Lcy*	Phatr3_J8835
Phytoene synthase	*Psy*	Phatr3_EG02349
Lutein deficient 1-like protein	*Lut 1*	Phatr3_J16586

**Table 2 antioxidants-09-00757-t002:** Carotenoid content (pg/cell) in *P. tricornutum* wild type (PT WT) and transgenic (T1, T2, T3) strains. Data represent mean (*n* = 3) and SD.

Strain	Fucoxanthin (pg/Cell)	β-Carotene (pg/Cell)	Diadinoxanthin (pg/Cell)	Diatoxanthin (pg/Cell)
PT WT	0.031 ± 0.001	0.0036 ± 0.0001	0.018 ± 0.001	0.0022 ± 0.0001
T1	0.036 ± 0.002	0.0042 ± 0.0001	0.024 ± 0.003	0.0032 ± 0.0003
T2	0.135 ± 0.004	0.0077 ± 0.0002	0.056 ± 0.002	0.0072 ± 0.0004
T3	0.092 ± 0.002	0.0086 ± 0.0003	0.059 ± 0.002	0.0076 ± 0.0003

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
