# Peer review of "Engineering the Unicellular Alga Phaeodactylum tricornutum for Enhancing Carotenoid Production"

_antioxidants, 2020, doi:10.3390/antiox9080757_

Round 1

Reviewer 1 Report

In this paper, the authors showed that the carotenoid production was enhanced by the overexpression of three genes in P. tricornutum. These findings are useful for the readers of Antioxidants. However, their discussion was not enough about the functions of VDE and VDR, especially VDE. VDE catalyzes the reaction from violaxanthin to zeaxanthin as you mentioned. So, OE of this enzyme is predicted to cause in the decrease of violaxanthin and consequently the decrease of fucoxanthin and diatoxanthin. However, the contents of those carotenoids were increased. If you have any idea, please discuss.

I recommend the publication after minor revisions.

Another comments are below.

  1. The words `P. tricornutum` were not sometimes italic (e.g. line158 and line159). Please check.
  2. In the panel c of figure 3, it was hard to see the signal for T3. If possible, you had better change the photo.
  3. In supplementary table S3, the values of the ‘ZEP1 + ZEP3` and ‘VDR + VDE + ZEP3’ were wrong? For example, #T3 of ‘VDR + VDE + ZEP3’ showed that the carotenoid sum (0.0702) was lower than fucoxanthin (0.0961) and diadinoxanthin + diatoxanthin (0.1826). Please check.

Author Response

In this paper, the authors showed that the carotenoid production was enhanced by the overexpression of three genes in P. tricornutum. These findings are useful for the readers of Antioxidants. However, their discussion was not enough about the functions of VDE and VDR, especially VDE. VDE catalyzes the reaction from violaxanthin to zeaxanthin as you mentioned. So, OE of this enzyme is predicted to cause in the decrease of violaxanthin and consequently the decrease of fucoxanthin and diatoxanthin. However, the contents of those carotenoids were increased. If you have any idea, please discuss.

We thank the reviewer for the appreciation of our work.

As far as the function of VDE and VDR, it is correct that intuitively overexpression of these enzymes should lead to less fucoxanthin. In the triple mutants however also more ZEP3 is present. In fact, we seem to have induced a change by overexpressing enzymes catalyzing opposite reactions, although the exact position of each enzyme in the pathway is not completely clear. We do not have a definitive explanation for the phenotype which we hypothesize could be due to a general perturbation of regulatory loops acting in the pathway, as mentioned on lines 359-361 (368-370 in the latest version). We believe that more detailed studies would be needed, for instance testing different light intensities and including the analysis of more single and double OE strains, for instance of the double Vdr/Vde OE strain mentioned at line 351 (362). This will be the subject of future studies where we hope to address questions on the role of each enzyme.

We have improved the sentence in the discussion at lines 380-383 (388-391).

I recommend the publication after minor revisions.

Another comments are below.

  1. The words `P. tricornutum` were not sometimes italic (e.g. line158 and line159). Please check.

This has been corrected.

  1. In the panel c of figure 3, it was hard to see the signal for T3. If possible, you had better change the photo.

We repeated the PCR and changed panel c in figure 3.

We had missed to specify how the colony PCR was made and have added a reference in the Methods (lines 181-182) (178-179).

  1. In supplementary table S3, the values of the ‘ZEP1 + ZEP3` and ‘VDR + VDE + ZEP3’ were wrong? For example, #T3 of ‘VDR + VDE + ZEP3’ showed that the carotenoid sum (0.0702) was lower than fucoxanthin (0.0961) and diadinoxanthin + diatoxanthin (0.1826). Please check.

We are sorry about this mistake, we had inadvertently inverted the third and the fourth columns for samples #I1 to #T19. We corrected the mistake and modified the table so that the total carotenoid (carotenoid sum) is now the last column for all samples.

Reviewer 2 Report

I have some minor comments regarding this manuscript, in addition to a few of typos and spellings marked in the pdf file.

The first question is the maximum increase on carotenoid levels relative to the wild type species. In the lines 234-235 you mention that only modified cells with a 2 fold relative increase were carried forward. What is the success rate of the biolistic technique you applied? This is an interesting information for the readers

Another interesting point is whether the gene integration was fixed in the subsequent generations and whether they showed identical characteristics and behaviour than the wild species.

Author Response

Reviewer #2

I have some minor comments regarding this manuscript, in addition to a few of typos and spellings marked in the pdf file.

- The first question is the maximum increase on carotenoid levels relative to the wild type species. In the lines 234-235 you mention that only modified cells with a 2 fold relative increase were carried forward. What is the success rate of the biolistic technique you applied? This is an interesting information for the readers

From literature the efficiency of biolistic transformation in P. tricornutum can range between 10–100 transformants / 108 cells (Apt, K.E., Grossman, A.R. & Kroth-Pancic, P.G. Stable nuclear transformation of the diatom Phaeodactylum tricornutum. Molec. Gen. Genet. 252, 572–579 (1996). https://doi.org/10.1007/BF02172403).

We performed around 57 different transformation experiments and found numbers of colonies compatible with expected efficiency in most of the cases, with some exceptions where the transformation was very inefficient or did not work at all (see column 2 in Supplementary table S2). This can happen and can be linked to many factors, including variability in the quality of the starting culture or of the plasmid preparations. We did not have the possibility to repeat all failed/inefficient experiments for practical constraints.

When many colonies appeared, we picked and screened a selection of them (see column 3 in Supplementary table S2). The fact that not all resistant colonies were positive (compare columns 3 and 4 in Supplementary table S2) depends on the co-transfection efficiency which is not always 100%, in some cases only the vector with the resistance cassette and not the vector/s for carotenoid genes overexpression had integrated. Moreover, positive double and triple transformants had to integrate three and four constructs, respectively (see lines 176-179)(172-176), therefore it is not unexpected to obtain a limited number of clones. Next, we choose to carry forward to the HPLC only liquid cultures that appeared to grow as well as the wild type. Finally, there is a degree of error in the HLPC large screening measurements and reliable results can be obtained using replicates, in our case it was not feasible to grow each positive culture in triplicate. We used a 2-fold change filter and only retained cultures with the most promising change in subsequent analyses.

We realized that the text was not clear and have made changes at lines 252-256 (269-272).

As to what number of strains with pigment change was to be expected, in the other similar studies we found the following figures:

- Two out of 66 psy transformants displayed a change in total carotenoid (Kadono et al., 2015).

- Three positive dxs and four positive psy transformants were analyzed for their carotenoid content and two of these displayed fucoxanthin increase. The total number of transformants obtained however was not provided (Eilers et al., 2016).

Therefore we think that our study did not suffer from any specific technical issue and is in line with similar studies.

While looking for these details in the relative papers we noticed some inconsistencies in the discussion at lines 317-321 (334-337) and corrected the statements and references.

- Another interesting point is whether the gene integration was fixed in the subsequent generations and whether they showed identical characteristics and behaviour than the wild species.

Biolistic transformation causes integration of the transgene in the genome, and generally results in the generation of stable transformants, that generally maintain the transgene over time. In this work, we have tested the same strain more than once over months, or even after more than a year, for the presence of the transgene by PCR or for the change in carotenoids with HPLC (see for instance Supplementary table S3 and lines 262-263(277-278)) and found reproducible results, indicating that the integration was fixed, as added at lines 332-334(345-347). In terms of characteristics and behaviour, we did not notice any obvious difference compared to the wild type as far as growth, color, density or morphology, however more in-depth analyses are needed to define whether increased pigment content leads to any phenotypic difference.

We made changes in the text for this point (lines 297-298(312-313) and 380-383(388-391)).

Replies to the points marked in the pdf, line numbers here refer to the first version:

Mistakes, typos and format style at lines at lines 57,110,129, 158, 159, 167, 180, 183, 212, 214, 220, 221 and 311 have been corrected.

Lines 94 and 95, there is no univocal convention for gene nomenclature in diatoms, here we have followed the convention for which the gene name should be capitalized and in italics and the protein name should be all in upper case, as also done in Lavaud et al., 2012 and Eilers et al 2016.

Line 332, we have substituted “personal communication…” with two relevant citations (Sachindra et al 2007 and Mikamo et al 2013).

Reviewer 3 Report

The manuscript of Manfellotto et al. is very interesting for the scientific community working on microalgae and their valuable compounds. The work, carried out with scientific rigor, it is well described and the literature cited properly. I want just to point out the need to consider carefully the effect of light. Generic assessment high light, normal light, low light, stressful lighting conditions are non-sense. Actually, different microalgae respond differently to varying light intensities depending on their genome but also to the acclimation to a defined environment. Thus, for example, a light intensity of 100 mmol m-2s-1, can result as high light or stressful conditions for a strain, but as low light conditions for other strains/species acclimated to higher light intensity. For this reason, I suggest specifying the intensity of the light instead of using HL, NL, LL (lines 293, 311, 314).

Another point is why the authors do not evaluate the response of the OE strains to higher light intensity in comparison to the 90 mmol m-2s-1 used for growth? High light conditions could enhance the pigment production in the OE strains.

Author Response

Reviewer #3

The manuscript of Manfellotto et al. is very interesting for the scientific community working on microalgae and their valuable compounds. The work, carried out with scientific rigor, it is well described and the literature cited properly. I want just to point out the need to consider carefully the effect of light. Generic assessment high light, normal light, low light, stressful lighting conditions are non-sense. Actually, different microalgae respond differently to varying light intensities depending on their genome but also to the acclimation to a defined environment. Thus, for example, a light intensity of 100 mmol m-2s-1, can result as high light or stressful conditions for a strain, but as low light conditions for other strains/species acclimated to higher light intensity. For this reason, I suggest specifying the intensity of the light instead of using HL, NL, LL (lines 293, 311, 314).

We thank the reviewer for the positive evaluation of our manuscript.

We also thank him/her for this pertinent comment, indeed P. tricornutum has a high acclimation capacity and has efficient growth rates under a wide range of light intensities. Based on ours and others’ experience, cells maintained at 90 μmol m−2 s−1 grow well and we consider this non-stressful.

We agree with the reviewer's opinion and changed the text accordingly. We added the light intensity values applied (line 137 (139) and elsewhere), and described more extensively the experiments done in some of the studies cited (lines 339-346 (350-357) and 353-357 (364-366)).

- Another point is why the authors do not evaluate the response of the OE strains to higher light intensity in comparison to the 90 mmol m-2s-1 used for growth? High light conditions could enhance the pigment production in the OE strains.

This is a pertinent suggestion, we could not test high light but have added a reference to this possibility in a sentence at the end of the discussion (lines 380-383 (388-391)).

Replies to the points marked in the pdf, line numbers here refer to the first version:

Mistakes, typos and format style at lines 38, 158, 180, 183, 245 and 327 have been corrected.

Line 222, we added more information in the Methods.

Detail has been added at line 232.

Line 266, “in panel b the stronger expression is not so evident for T3 but only for T2”.

Yes this is correct, changes in T2 are stronger than in T3, as evident also in panel c in the same figure.

Lanes reported in figure 4 b refer to two different membranes (one for HA and one for control detection) and adjustments were made in the brightness of the control membrane when performing the densitometry measurements in order to be able to calculate a fold change.

In any case we repeated the intensity measurements, using the Quantity-One software, on the original acquisition that we used to make the panel in figure 4, the densitometry raw data are reported in the table below, where the difference are consistent with data shown in the figure.

VDR-HA

ZEP3-HA

VDE-HA

T1

7.772.409

37.828.278

28.591.467

T2

19.693.196

67.170.642

56.002.538

T3

14.793.309

39.655.884

37.227.395

At line 318, “Are the high light conditions reported in the two papers cited higher than those reported here?”, the answer is yes, HL is 700 μmol m−2 s−1 in Kuczynska et al 2020 and 500 μmol m−2 s−1 Nymark et al 2009, as indicated on lines 340-346, and we used 90 μmol m−2 s−1, as stated in the Methods.

322-323, yes we recognized that the sentence was not clear and have modified the text.